# Microglial activity during postnatal development is required for infantile amnesia in mice

Erika Stewart[1,2], Louisa G. Zielke[1,2], Antje R. de Boer[1,2], Gabrielle Guillaume Boulaire[1,2], Sarah D. Power[1,2,3], Tomás J. Ryan [1,2,4,5]*

1 School of Biochemistry and Immunology, Trinity College Dublin, Dublin, Ireland, 2 Trinity College Institute for Neuroscience, Trinity College Dublin, Dublin, Ireland, 3 Center for Lifespan Psychology, Max Planck Institute for Human Development, Berlin, Germany, 4 Florey Institute of Neuroscience and Mental Health, University of Melbourne, Melbourne, Victoria, Australia, 5 Child & Brain Development Program, Canadian Institute for Advanced Research (CIFAR), Toronto, Ontario, Canada

* tomas.ryan@tcd.ie

## Abstract

Infantile amnesia, the inability to recall episodic memories formed during early childhood, is a hallmark of postnatal brain development. Yet the underlying mechanisms remain poorly understood. This work aimed to gain a better mechanistic understanding of infantile amnesia. Microglia, specialized macrophages of the central nervous system, are known to play an important role in synaptic refinement during postnatal development and have recently been implicated in memory-related functions. Using mouse models, we identified microglia as key regulators of memory accessibility in infancy. We profiled dynamic changes in microglial morphology across the postnatal window that paralleled the onset of infantile forgetting. We found that pharmacological inhibition of microglial activity during a specific postnatal window prevents infantile amnesia for a contextual fear memory, implicating microglia as active modulators of infant memory persistence. Using activity-dependent tagging of infant encoded engram cells, we demonstrated that microglial inhibition alters engram size and engram reactivation in the amygdala and results in changes in microglia–engram cell interactions. Furthermore, we characterized a relationship between microglial dysfunction and the lack of infantile amnesia in maternal immune activation offspring. Together, these findings reveal a novel role for microglia in regulating infant memory retrieval in mice and suggest that microglial dysfunction may contribute to altered memory trajectories in neurodevelopmental disorders.

## Introduction

Infancy and childhood are developmental periods rich in novel experiences and stimuli as children actively explore the environment and assimilate new information to build internal representations of the world [1]. Yet, human infants rapidly forget early episodic and contextual memories, a phenomenon known as infantile amnesia [2–4].

**Data availability statement:** All data necessary for the interpretation and replication of the reported findings are contained within the manuscript and supplemental information. Statistical source data is provided with this paper. Detailed figure source data is also provided with this paper.

**Funding:** This work was funded by the Lister Institute of Preventive Medicine (https://lister-institute.org.uk/, Fellowship to T.J.R.), European Research Council (https://erc.europa.eu/; 715968 to T.J.R.), Irish Research Council (https://research.ie; GOIPG/2020/1294 to E.S.), Research Ireland (https://www.researchireland.ie/; 15/YI/3187 and 24/FFP-A/13397 to T.J.R.), the Jacobs Foundation (https://jacobsfoundation.org/; 2019-1356-2 to T.J.R.), the Canadian Institute for Advanced Research (https://cifar.ca/); CF-0303 to T.J.R.), and the US National Institute of Health (https://www.nih.gov/; 1R01NS121316 to T.J.R.). The funders had no role in study design, data collection and analysis, decision to publish, or preparation of the manuscript.

**Competing interests:** The authors have declared that no competing interests exist.

**Abbreviations:** AD, Alzheimer's disease; AMG, amygdala; ASD, autism-spectrum disorder; BLA, basolateral amygdala; CeA, central amygdalar nucleus; CFC, contextual fear conditioning; DG, dentate gyrus; ECM, extracellular matrix; HFD, High-fat diet; HPRA, Health Products Regulatory Authority; MIA, maternal immune activation; PBS, phosphate-buffered saline; PFA, paraformaldehyde; PNNs, perineuronal nets; RSC, retrosplenial cortex; SEM, Standard errors of the mean; STED, stimulated emission depletion; TBI, traumatic brain injury; TRAP2, Targeted Recombination In Active Populations; 4-OHT, 4-hydroxytamoxifen.

This form of rapid forgetting is conserved across species, is well characterized in rodent models, and occurs during a period of robust brain development [5]. However, while the behavioral features of infantile amnesia have been well characterized, the underlying cellular and molecular mechanisms that drive it remain poorly understood [5–8]. Understanding how the brain regulates memory accessibility during early development is critical for uncovering fundamental principles of memory formation and for identifying mechanisms that may be disrupted in neurodevelopmental disorders [9]. However, the cellular substrates of infantile forgetting, and how they intersect with key developmental processes, are largely unknown.

Memory is believed to be stored in specific cell ensembles known as engrams, that are active at the time of learning and reactivated during recall, to drive memory-specific behaviors [10–12]. Through the use of activity-dependent labeling we can investigate changes in engram cells across the life span [13]. However, how engram properties evolve during infancy, and what mechanisms render them inaccessible during infantile amnesia, has been scantly explored. By focusing on key developmental processes occurring during this period of early brain development, we aimed to identify potential mediators of infantile amnesia.

Human epidemiological studies highlight a link between gestational inflammation and risk of neurodevelopmental disorders [14,15]. Animal models of maternal immune activation (MIA) show altered brain development and behavioral deficits reminiscent of autism-spectrum disorder (ASD) and schizophrenia [16–20]. We previously described a phenomenon in which MIA prevents infantile amnesia in male offspring [21]. Using this MIA model as a guide, we focused on potential mechanisms that may be mediating this change in infant memory. MIA is known to modulate the activity of microglia in the brains of offspring [22–24]. Thus, we hypothesized that microglia are one of the candidate mediators for the modulation of infant memory.

Microglia are specialized macrophages that reside in the central nervous system and contribute to brain development, plasticity, and neurodegeneration [25–27]. Microglial cell numbers steadily increase during early postnatal development before stabilizing during the third postnatal week [28]. Additionally, this developmental window is defined by ongoing synaptic maturation and elimination where microglia are key players in this early synaptic refinement [29–31]. Other work has described microglial regulation of neuronal activity and learning-dependent synaptic remodeling [32,33].

In the present study, we examined the role of microglia in infantile amnesia. We first characterized changes in microglial morphology across the infantile window as proxy for changes in microglial activity. We used pharmacological and receptor-specific inhibition approaches and tested whether microglial inhibition alters memory persistence, engram dynamics, and microglia-engram interactions. Finally, we asked whether modulating microglial function in MIA offspring could reinstate infantile amnesia. Our findings revealed a previously unrecognized role for microglia in developmental memory regulation and suggest that microglial dysfunction may contribute to altered memory outcomes in neurodevelopmental disorders in mice.

## Results

### Changes in microglia activity across infant development mirrors memory retention

To assess memory retention in mice we employed a classical Pavlovian contextual fear conditioning (CFC) paradigm (Fig 1A). Infant mice were trained on postnatal day 17 (P17) and tested for recall in the same context either 3, 5, or 8 days later on P20, P22, and P25, respectively (Fig 1A). Adult mice were trained between 8 and 9 weeks of age (P56) and similarly underwent a recall test either 3, 5, or 8 days later (Fig 1A). We first quantified the baseline and shock responses of both infant and adult mice during training (Fig 1B). Infant and adult mice showed equivalent baseline levels of freezing, but infants demonstrated a higher response to the first shock with overall comparable levels between ages (Fig 1B). We then quantified freezing behavior during recall as a readout of fear memory (Fig 1C and 1D). Consistent with previous findings, infant mice exhibited a graded decrease in the levels of freezing across this developmental window, with high levels of freezing observed at both P20, reduced levels at P22 before the onset of amnesia at P25 (Fig 1C) [8]. In contrast, adult mice displayed stable memory retention demonstrated by high levels of freezing at all time points (Fig 1D). To investigate morphological changes, subjects were sacrificed, perfused, and brains were collected for histological analysis (Fig 1E). Morphometric profiling of microglia has been classically used as a proxy of microglial activation state in addition to maturation [34,35]. Here, we characterized microglial morphology across the infantile amnesia window following CFC to identify changes in microglia that may coincide with the onset of infantile amnesia. We focused specifically on two brain regions of interest; the hippocampal dentate gyrus (DG) which plays a critical role in supporting the encoding and retrieval of contextual information [36] and the amygdala (AMG); an integrative hub that plays a key role in processing emotional information [37,38]. Both areas are recruited during contextual fear memory formation [39].

We stained brain sections for a well-established marker of microglia, ionized calcium-binding adapter molecule 1 (Iba1) (Fig 1F and 1G; top). We performed surface rendering and 3D reconstruction of Iba1+ labeled cells in the DG and AMG and we quantified numerous morphometric parameters of microglia at these different time points (Fig 1F, and 1G; bottom). In the DG, we observed a significant difference in microglial branching and filament length between 3 (P20) and 8 days (P25) (Fig 1H–1J). In the AMG, we observed significant differences in branching, the number of terminal points and filament length between 3 (P20) and 5 days (P22) (Fig 1K–1M). In adults, measures of microglial morphology in DG remained consistent (Fig 1N–1P). Interestingly, in the adult AMG, we observed significant differences in the number of branch points and terminal points between the 3 and 8-day recall group suggesting some changes in microglia following adult recall at 8 days post-training (Fig 1Q–1S). The observed changes in microglial morphology follows a pattern of change that mirrors memory retention in both infants and adults. We also note that the differences in quantity of morphological changes between adults and infants likely reflect the relative levels of cell maturity [34,40]. To further explore changes in microglial state across the infantile window, we quantified the expression of CD68, a marker of phagolysosomes often used as a proxy of microglial activity and phagocytosis, at 3, 5, and 8 d post-CFC (Fig 1T). We observed changes in microglial CD68 expression within microglial cells that also mirrored infant memory retention (Fig 1U and 1V). These findings highlight microglia as a prime candidate for further mechanistic investigation in the context of infantile amnesia and memory.

### Pharmacological inhibition of microglial activity prevents infantile amnesia

Given the changes we observed in microglial morphology and activation state across the infantile window (Fig 1), in addition to their well-described developmental roles and their implication in memory modulation, we hypothesized that microglial activity drives infantile amnesia. To test this, we treated mice postnatally with minocycline hydrochloride, a second-generation tetracycline antibiotic widely used to inhibit microglial activity [41–44]. Minocycline was administered either via drinking water from P16 to P25 (Fig 2A) or through daily intraperitoneal (i.p) injections (S1A Fig). Control groups received regular drinking water or saline injections, respectively. Mice were trained at P17 on a CFC paradigm in context

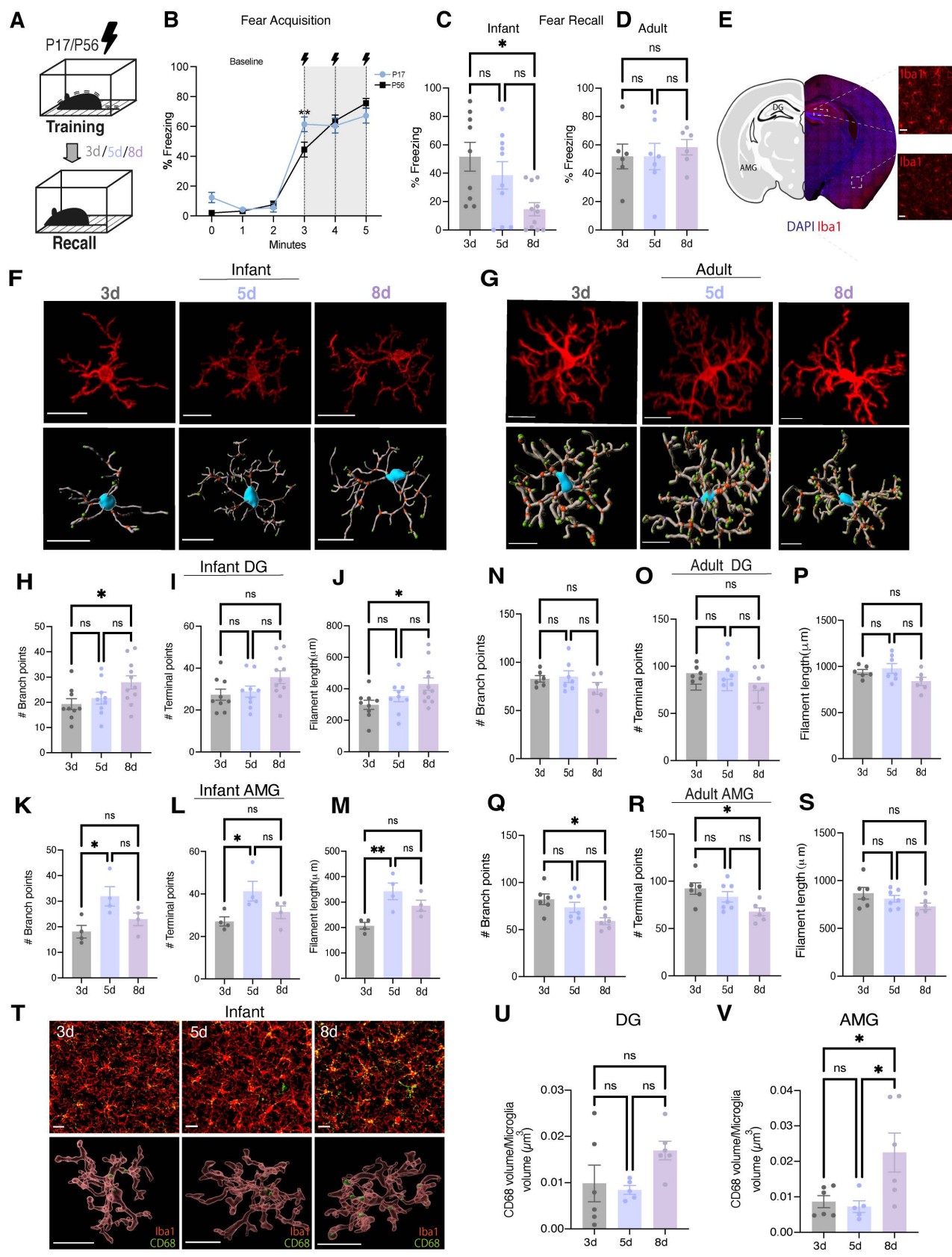

PLOS Biology

**Fig 1. Microglial morphology and CD68 expression dynamically changes in the hippocampal DG and amygdala across the infantile window.** **(A)** Schematic diagram of experimental schedule. Infant mice were trained on P17 on a CFC task and underwent a recall test either 3, 5, or 8 days later before being sacrificed for tissue collection. Adult mice underwent the same CFC and underwent a recall test either 3, 5, or 8 days later before being sacrificed for tissue collection. **(B)** Quantification of initial shock responses during CFC. **(C, D)** Quantification of freezing behavior during recall tests. **(E)** Iba1+ cells in DG and AMG. *Created in BioRender. Stewart, E. (2025)* https://BioRender.com/hf3ttgz. **(F, G)** Representative images of Iba1+ stained images and 3D filament reconstructions of microglia. Scale bar, 20 μm. **(H–M)** Quantification of microglial morphology in infant mice. **(N–S)** Quantification of morphological measures in adult mice. **(T)** Representative image IBA1 and C68 staining and 3D rendering of CD68 expression within microglia. Scale bar 30 μm (upper panel) and 20 μm (lower panel). **(U, V)** Quantification of CD68 volume within Iba1+ cells in infants DG and AMG. Black lightning symbol represents foot-shocks. Data is presented as mean±SEM. Each data point represents individual mice. $n = 4$–11 mice/group. 2 region of interest (ROI) images were analyzed per mouse from either the DG molecular layer or BLA. 3–5 microglia were analyzed per 2 ROI image. Individual cells were pooled per mouse for statistical comparisons. One-way ANOVA with Tukeys post hoc. n.s $P > 0.05$, $*P < 0.05$, $**P < 0.01$. Details of all statistical comparisons may be found in S1 Data. The data underlying this Figure can be found in S2 Data.

A and tested for memory recall 1 day later (P18) in the same context to confirm memory formation and 8 days later (P25) to confirm infantile amnesia (Figs 2B and S1B). Both control and minocycline-treated mice displayed equivalent levels of freezing 1-day post-training demonstrating intact fear memory (Figs 2C and S1C). However, 8 days post-training, the minocycline group displayed significantly higher levels of freezing compared to control-treated group (Figs 2C and S1C). The high level of freezing exhibited by the minocycline-treated group at P25 indicates preserved fear memory despite the typical onset of infantile amnesia, suggesting that microglial inhibition prevents infantile amnesia. To verify memory specificity, we tested mice in novel context B. Minocycline-treated mice did not exhibit higher levels of freezing compared to controls indicating context-specific memory recall rather than a generalized freezing phenotype resulting from minocycline treatment (S1C Fig). As an additional control, we demonstrated that minocycline-treated mice, that did not receive a foot shock during training do not freeze in Context A, again confirming that this freezing behavior is as a result of fear memory recall (S1F Fig). We did not observe any effect of sex on the effect of minocycline treatment on infantile amnesia (S1I Fig). Both male and female mice treated with minocycline showed significantly higher freezing levels 8 days post-training compared to controls (S1I Fig).

To confirm that pharmacological treatment with minocycline was indeed influencing microglial activation, mice were euthanized and perfused following recall 8 days post-training and brain tissue was collected. Tissue was stained for Iba1 and CD68 (Fig 2D; top). We carried out 3D surface rendering of microglial cells and CD68 protein expression within the DG (Fig 2D; bottom). We found significantly less CD68 expression within Iba1+ cells in the hippocampus of subjects treated with minocycline compared to controls (Fig 2E).

The infantile period is characterized by ongoing developmental mechanisms that drive critical period plasticity [45]. One of these mechanisms is the stabilization of inhibitory tone and the maturation of extracellular matrix (ECM) structures knowns as perineuronal nets (PNNs) [46,47]. Here, we found that minocycline-treated mice had significantly less PNNs in the hippocampus, specifically in the DG and CA1 compared to controls (S2 Fig).

To substantiate the above results, we used a more specific inhibitor of microglial-neuronal communication. Microglial activity and interaction with neurons are modulated by numerous factors, including soluble factors known as chemokines [48]. The chemokine CX3CL1 (fractalkine) is expressed by neurons and binds to its target receptor CX3CR1 that is expressed by microglia and is an important mediator of neuronal-glial communication that has been reported to play a crucial role in postnatal brain development [29,49]. We sought to investigate the potential role of CX3CL1-CX3CR1 signaling on infantile amnesia, using JMS-17-2, a potent and selective antagonist of CX3CR1 [44]. Infant mice were treated with JMS-17-2 or control through daily i.p injections from P16 to P25 (Fig 2F). Mice were trained on P17 in Context A and tested 8 days later at P25 (Fig 2G). Mice treated with JMS-17-2 displayed significantly higher levels of freezing 8 days post-training compared to control-treated mice (Fig 2G). To confirm that JMS-17-2 by itself or repeated daily i.p injections did not cause increased generalized anxiety or freezing, we compared mice that received CFC and JMS-17-2 (JMS-S) treatment with mice that received only a neutral contextual exposure

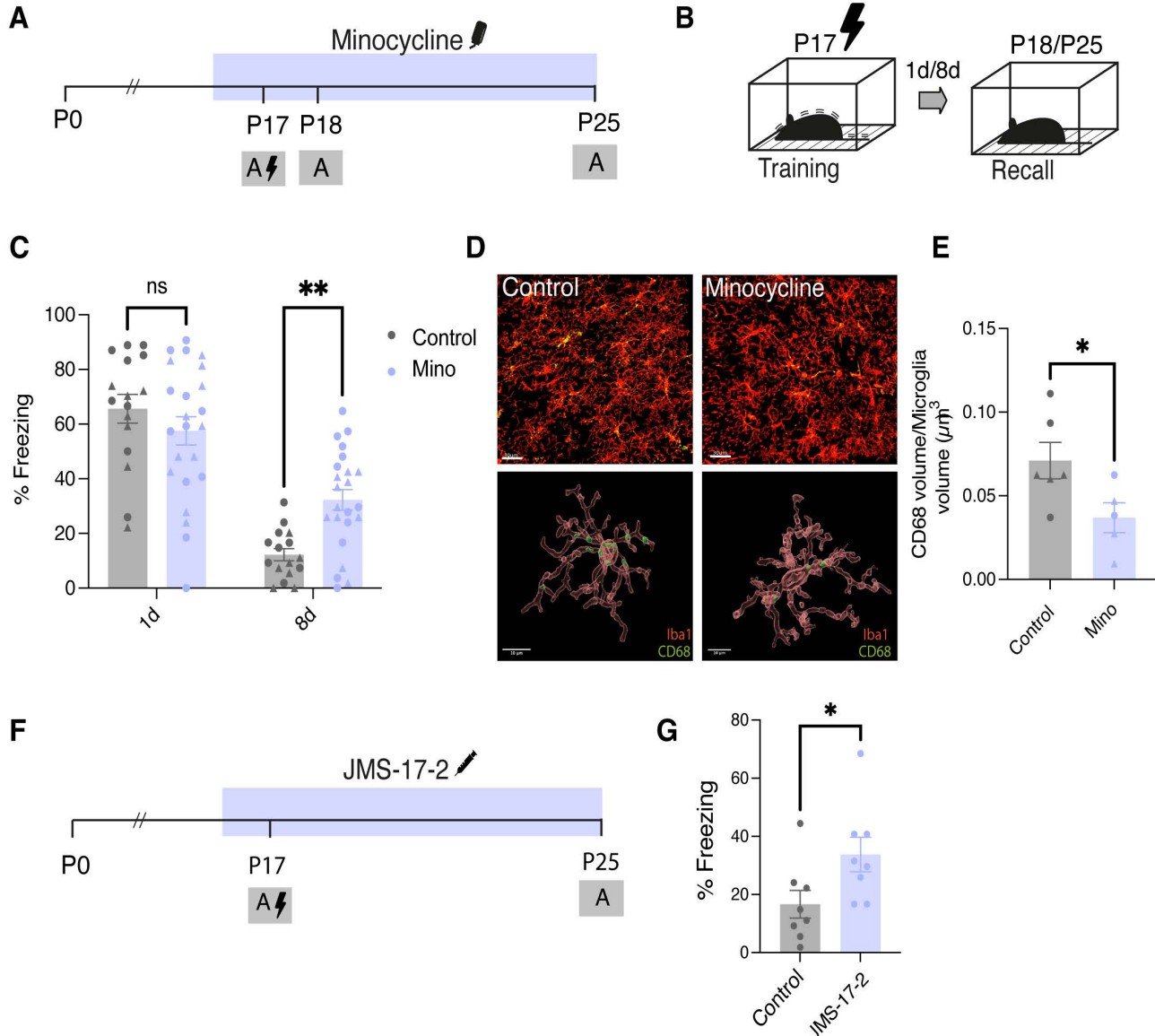

**Fig 2. Postnatal administration of minocycline prevents infantile amnesia for fear memory. (A)** Schematic diagram experimental schedule. Mice were administered minocycline (50 mg/kg) through drinking water from P16-P25. **(B)** Schematic of behavioral schedule. Male and female mice were trained at P17 and underwent recall test 1 and 8 days post-training. **(C)** Quantification of freezing behavior of minocycline or control mice 1 and 8 days post-training. **(D)** Representative image of 3D rendering of CD68 expression within microglia. Scale bar 30 μm (upper panel) 10 μm (lower panel). **(E)** Quantification of CD68 volume within Iba1+ cells. **(F)** Schematic diagram experimental schedule. Mice were administered JMS-17-2 (10 mg/kg) through i.p injection from P16-P25. **(G)** Quantification of freezing behavior of control of JMS-treated mice 8 days post-training. Black lightening symbol represents foot-shocks. Males are represented as circle symbol and females as triangle symbol on graphs. Data is presented as mean±SEM. **(C)** 21 mice per group. **(E)** Each point represents an individual mouse. $n = 5$–6 mice/group. 4–5 microglia were analyzed from 2 ROI images per mouse. Individual cells pooled per mouse for statistical comparisons. **(G)** $n = 8/9$ mice per group. Statistical comparison performed using **(C)** RM Two-way Anova with Bonferroni **(E, G)** Students $t$ test *$P < 0.05$, **$P < 0.01$. Details of all statistical comparisons may be found in S1 Data. The data underlying this Figure can be found in S2 Data.

with no foot-shock but also received JMS-17-2 treatment (JMS-NS) (S1J Fig). The lack of freezing in the JMS-NS group shown in S1L Fig indicates that freezing is due to retention of the fear memory. These results support our hypothesis that microglial activity throughout postnatal development may contribute to infantile amnesia potentially mediated by CX3CL1-CX3CR1 signaling.

## Inhibition of microglial activity alters infant engram dynamics

Given that microglial inhibition preserved memory retention in infant mice past the onset of infantile amnesia, we next asked whether these behavioral effects were associated with changes in memory engram activity. We utilized a transgenic Cre-based engram tagging strategy to label infant engram cells and evaluate engram dynamics. We crossed a Targeted Recombination In Active Populations (TRAP2) mouse, in which the *c-fos* promoter drives expression of tamoxifen-inducible Cre recombinase (iCre), with an Ai32 reporter line expressing channelrhodopsin/enhance yellow fluorescent protein (Chr2-EYFP) [50,51]. This allowed for permanent tagging of infant engram cells with EYFP (Fig 3A). We hypothesized that microglial inhibition would lead to increased infant engram reactivation. To test this, we administered minocycline in drinking water while controls received regular drinking water. Mice underwent CFC at P17, receiving an injection of 4-OHT immediately following training to label active cells (Fig 3B). Mice were tested 8 days later and perfused 45 min following a recall test for tissue collection (Fig 3B).

We quantified engram reactivation in the hippocampal DG, the basolateral amygdala (BLA), the central amygdalar nucleus (CeA), and the retrosplenial cortex (Figs 3C–3G and S4). For each region, we quantified engram size (% EYFP+ cells), the number of active cells (% c-Fos+ cells), and engram reactivation (% double positive EYFP+ and c-Fos+ cells) at the time of recall in each region. In the DG and RSC, no differences were observed between groups in the number of EYFP+ cells, number of c-Fos+ cells, or overlap (Fig 3H–3J, 3Q-3S). However, in the BLA and CeA, minocycline-treated mice exhibited significantly increased engram reactivation compared to controls (Figs 3M, 3P, and S4D), consistent with our previous findings that artificial reactivation in the DG of an infant encoded engram post-amnesia also results in increased BLA activation (Figs 3M and S4) [21]. Unexpectedly, we also found a difference in the number of engram cells in the BLA of minocycline-treated mice (Fig 3K). As a control experiment, we also evaluated engram reactivation following minocycline treatment in mice that received CFC (shock) and those that received only a contextual experience without shock (NS) (S4 Fig). We observed significant differences in levels of reactivation in both the DG and BLA of S and NS minocycline-treated mice (S4 Fig).

Overall, these results indicate that minocycline treatment both increases engram size and reactivation in the AMG. These data further support our hypothesis that microglia regulate infantile memory through region-specific effects on engram activity.

Here, we found that minocycline treatment attenuated microglial activation, lead to the persistence of infant fear memory, and increased infant engram reactivation in the AMG. To further investigate the relationship between microglial plasticity and engram function, we quantified microglial morphology in the AMG following CFC and minocycline treatment (Fig 3T). Compared to controls, minocycline-treated mice exhibited increased microglial branching, with trends toward increased terminal points and filament length (Fig 3U–3W). To assess how these morphological features related to memory performance and engram activity, we correlated branching complexity with freezing behavior and engram reactivation (Fig 3X and 3Y). Both measures were positively associated with microglial branching (Fig 3Y), linking reduced microglia activation to enhanced memory recall and engram reactivation.

Microglia are highly sensitive to changes in neuronal activity and two-photon imaging of microglia reveals targeted interactions between microglial processes and dendritic spines [52]. We hypothesized that microglial-engram interactions may regulate engram accessibility. To test this, we evaluated microglial interactions with engram cells in the AMG following minocycline treatment. Using 3D surface rendering and reconstruction of EYFP+ engram dendrites and Iba1+ microglial cells in the AMG, we quantified the number of direct contacts between microglia processes and engram cells (Fig 3Z).

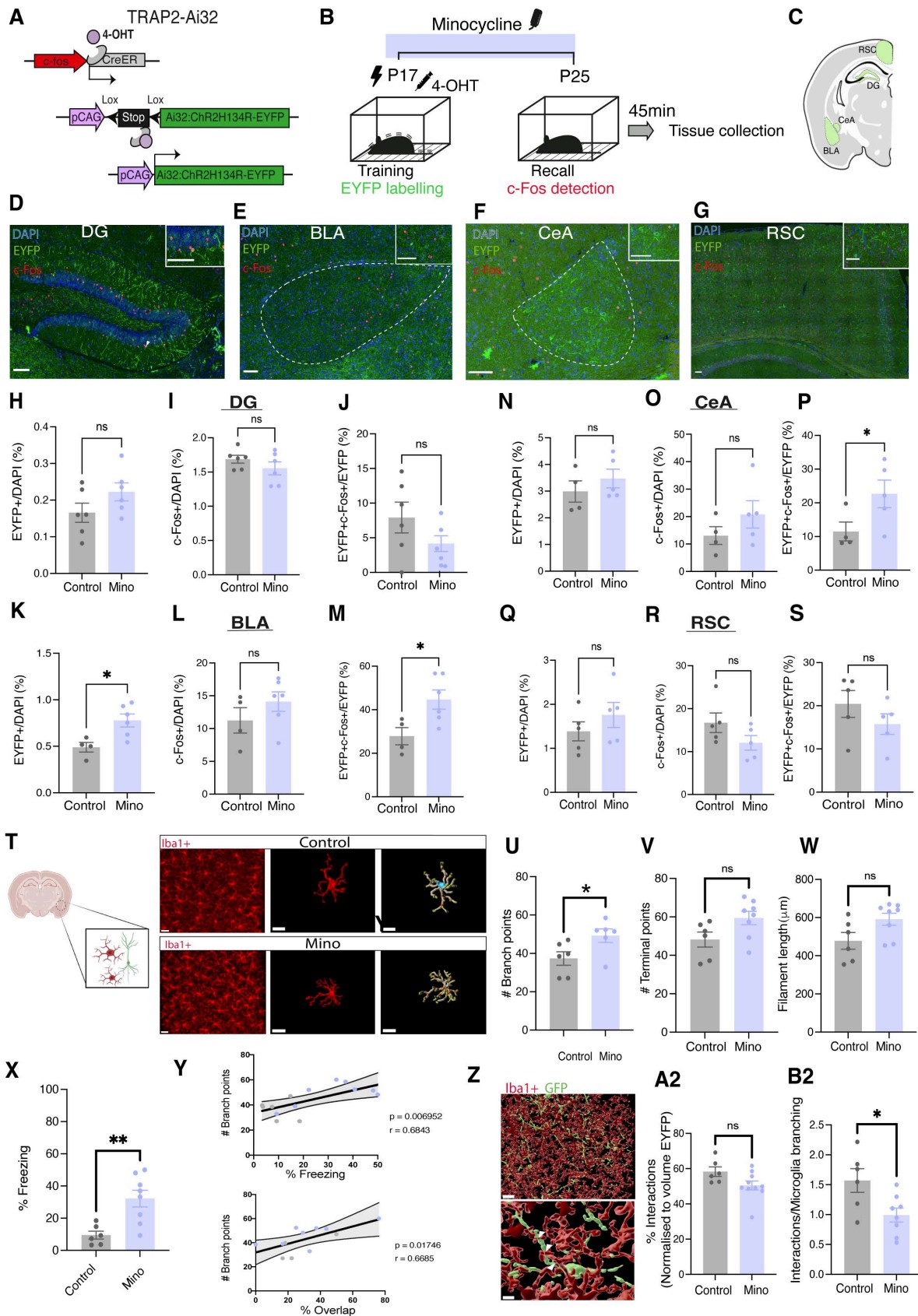

**Fig 3. Postnatal minocycline treatment increases engram size and reactivation in the amygdala. (A)** Diagram of tamoxifen-driven inducible genetic strategy for engram cell tagging. **(B)** Schematic diagram of experimental schedule. Mice were administered 50 mg/kg minocycline from P16-P25 through drinking water. Mice were trained at P17 and underwent a recall test 8 days later and were scarified 45 min following recall. **(C)** Created in BioRender. Stewart, E. (2025) https://BioRender.com/gq535e6. **(D)** Representative images of engram cell labeling (Green) and c-Fos+ cells (Red) and overlap in the DG, BLA, CeA, and RSC. Arrowhead indicates an example of overlapping cells. Large-scale bar, 150 μm. Small-scale bar, 75 μm. **(H–J)** Quantification of % Engram cells, % c-Fos, and engram overlap/reactivation in DG, **(K–M)** BLA, **(N–P)** CeA, and **(Q–S)** RSC. $N = 4$–6 litters/group, $n = 4$ slices per animal. **(T)** Representative images of Iba1+ microglial cells in the BLA and 3D reconstruction of individual microglial cells using IMARIS software. Scale bar 20 μm (left panel) 15 μm (middle and right panel). Created in BioRender. Stewart, E. (2025) https://BioRender.com/40c19e9. **(U–W)** Quantification of measures of morphological plasticity in microglia between control and minocycline-treated mice. **(X)** Quantification of freezing behavior during P25 recall test in T2-Ai32 mice treated with either minocycline or vehicle. **(Y)** Scatter-plot of the relationship between microglial branching and engram reactivation or freezing behavior. **(Z)** Representative image of microglia-engram interaction. Scale bar, 20 μm (top panel) 5 μm (lower pane). **(A2, B2)** Quantification of % interaction between microglial and engram surfaces. Syringe symbol represents 4-OHT injection. Black lightning symbol represents foot-shocks. Data is presented as mean ± SEM. For engram quantification, each point represents individual litter. Counts per slice pooled per litter for statistical comparisons. For microglial quantification, $n = 6$–8 mice, 8 microglia/mouse. For interaction analysis, 6–10 mice/group, 2/3 sections per mouse. **(U–W)** Statistical comparison performed using **(H–S)** Nested $t$ test, **(U–W)** Student $t$ test, **(X–Y)** Pearsons correlation; n.s $P > 0.05$, *$P < 0.05$. Details of all statistical comparisons may be found in S1 Data. The data underlying this Figure can be found in S2 Data.

We found that Minocycline-treated mice exhibit a significant reduction in the number of microglia-engram contact points (Fig 3B2), suggesting that decreased microglial engagement may facilitate engram stabilization.

## Microglia mediate the effect of MIA on infantile amnesia

Our previous work characterized a developmental brain state in which infantile amnesia does not occur [21]. Gestational immune activation through the delivery of a viral-mimetic polynosinic-polycytidylic acid (Poly(I:C)) at embryonic day 12.5 (E12.5) results in male offspring that retain memories acquired in infancy, bypassing the typical forgetting observed during this period [21]. Here, we also tested the effect of another maternal environmental intervention on infantile amnesia, to determine whether this prevention of infantile amnesia was specific to a timed inflammatory insult at E12.5 or if lower-grade chronic inflammation was sufficient to produce the same effect on infant memory (S5 Fig). Maternal HFD had no effect on infantile amnesia, which suggests that there may be a sensitive window for the effects of maternal inflammation (S5C Fig).

Our findings in Figs 2 and 3 describe a novel condition in which infantile amnesia is prevented through pharmacological inhibition of microglia. A common factor between these conditions and MIA, is their impact on microglia. Substantial evidence from both animal models of MIA and human samples from ASD or schizophrenia cases indicates a disruption to microglial function [24,53–56]. In line with other work, we found changes in embryonic microglial gene expression following MIA (S6 Fig). We observed changes in expression of some microglial-related genes in embryonic brains following MIA (S6 Fig). Inspired by work that targeted microglia to prevent or reverse behavioral deficits in MIA offspring, we then investigated the effect of microglial inhibition on infantile amnesia in our MIA offspring [22,23,57]. To test this, we inhibited microglial activity during the early postnatal period in MIA offspring. MIA offspring were treated with minocycline via drinking water from P0 to P14, a window selected to intervene prior to memory acquisition at P17 (Fig 4A). Control MIA offspring received regular drinking water. Our logic for targeting microglia in MIA offspring during this early postnatal window is based on previous work that has demonstrated premature shifts in the trajectory of microglia development and phenotype in MIA offspring [58]. Mice were trained on a CFC paradigm at P17 and tested 8 days later (Fig 4A). Minocycline-treated MIA offspring displayed significantly lower freezing levels compared to untreated MIA offspring (Fig 4B). Crucially, minocycline treatment had no effect on control offspring treated during the same developmental window of P0-P14 (S6E and S6F Fig). This result suggests that early postnatal microglial inhibition was sufficient to restore infantile amnesia in MIA offspring. Following a recall test at P25, mice were then either returned to their homecare to be evaluated in a battery of tests to assess for ASD phenotypic behaviors in adulthood or immediately perfused for histological analysis. Sociability was tested using the 3-chamber social interaction task and repetitive behavior was assessed using the marble

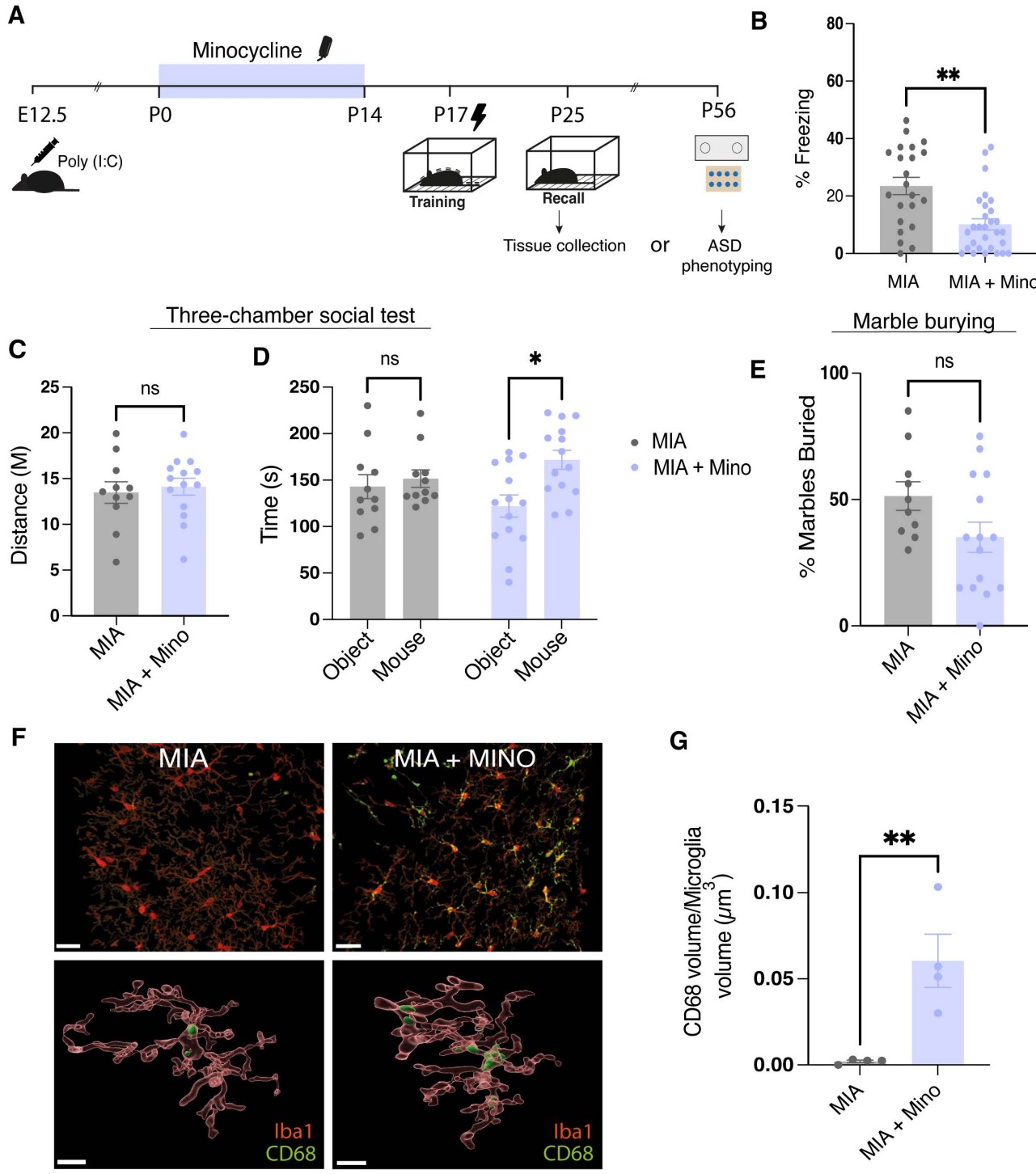

**Fig 4. Early-postnatal treatment of minocycline in MIA offspring prevents memory phenotype. (A)** Schematic diagram of experimental schedule. Pregnant dams received injection of Poly(I:C) at E12.5. Minocycline was administered from P0 to P14. Male offspring underwent CFC at P17 and were tested for memory recall 8 days later. **(B)** Quantification of freezing behavior of mice during recall test. **(C–E)** ASD phenotyping battery of behaviors. **(C)** Quantification of distance traveled in 3-chamber-test. **(D)** Time spent in the object or the mouse zone during test. **(E)** Quantification of % marbles buried. **(F)** Representative images of z-stacks and 3D reconstructions of microglial cells from MIA and MIA+Minocycline treated offspring, Scale bar 30 μm (upper panel), 10 μm (lower panel). Sections were stained with Iba1 (red) and CD68 (green). **(G)** Quantification of CD68 within Iba1+ cells. Data is

presented as mean±SEM. Points represent individual mice. For freezing behavior, $N=9$–$10$ litters/group, $n=22/29$ mice/group. For ASD phenotyping, $n=11$–$14$ mice/group. For microglia histology, each point represents individual microglial cell. $N=4$ mice/group, $n=8$ microglia cells per group. Statistical comparison performed using **(B)** Student nested $t$ test. **(C, E, G)** Student $t$ test. **(G)** Two-way ANOVA with Bonferroni post hoc; n.s $P>0.05$, *$P<0.05$, **$P<0.01$. Details of all statistical comparisons may be found in S1 Data. The data underlying this Figure can be found in S2 Data.

burying assay (Fig 4C–4E). Minocycline-treated MIA offspring demonstrated an increased preference for a social stimulus (a novel mouse) compared to a nonsocial object, whereas untreated control MIA offspring showed no preference (Fig 4D). We also observed a trend towards a decreased tendency for marble burying in minocycline-treated MIA offspring (Fig 4E).

Lastly, we evaluated the effect of MIA and early postnatal minocycline treatment on microglial activation (Fig 4F and 4G). In accordance with other reports, our histological analysis revealed that MIA results in decreased expression of CD68 in microglial cells suggesting a reduced phagocytic activity [23] (Fig 4). Interestingly, early postnatal minocycline treatment of MIA offspring resulted in a significantly higher level of CD68 expression at P25 (Fig 4G). Together, these data support a model in which MIA induces alterations in microglial function, disrupting the developmental processes that normally give rise to infantile amnesia.

## Discussion

The findings presented in this study demonstrate clear evidence for a role for microglia in infantile amnesia in mice. We characterized microglia morphology throughout the developmental window of interest and found that microglia exhibit changes in morphology and CD68 expression across this window of interest suggesting changes in their activity and/ or function. Through pharmacological inhibition of microglial activity, we have shown that microglial activity is necessary for infantile amnesia of a contextual fear memory to occur, and microglia may modulate engram activity through the microglia-neuronal axis via CX3CL1-CX3CR1 signaling. By tagging infant engram cells during learning, we further illustrate that microglial inhibition results in differences in engram size and reactivation specifically in the AMG. Our data aligns with existing evidence that microglia contribute to decreased fear expression in adult mice [43] but build on these findings by demonstrating a specific role for microglia in the rapid forgetting experienced throughout infancy, and delineating their impact on infant engram dynamics. Additionally, our data indicates that microglial dysfunction may underpin developmental and behavioral changes in MIA offspring and presents original evidence that links microglial activity with our previous published finding on changes in infant memory in MIA offspring [21].

Microglia are the central nervous system representatives of the immune system that mediate immune responses, promote injury repair, and fight invading pathogens [59]. Beyond their well-described immunological roles, microglial actively interact with neurons to modulate neuronal circuits with resulting impacts on cognition and behavior [60–62]. A role for microglial activity has been described in the regulation of numerous forms of behavior, for example, the extinction of anxiety-like behaviors or juvenile social and play behaviors [44,63]. Specifically, the synaptic modulatory functions of microglia position them as ideal plasticity managers [32,33]. Indirect evidence for a microglial role in memory primarily stems from their association with the pathogenesis of aging-related cognitive decline, traumatic brain injury (TBI) and Alzheimer's disease (AD), in which memory deficits are a defining feature [64–68]. However, recent studies have provided evidence for a direct role for microglia in a time-dependent decline in adult fear memory expression and have detailed specific microglia-engram interactions in the adult mouse brain [43,69]

Here, we provide the first evidence for a role of microglia in infantile amnesia. We show specific developmental changes in microglia activity following fear memory learning (Fig 1). We demonstrate that inhibiting microglial activity not only prevented infantile amnesia behaviorally (Fig 2) but also resulted in changes in engram activity (Fig 3). We observed enhanced reactivation of AMG engram cells in minocycline-treated mice (Fig 3K). These data indicate that changes in microglial activity influences engram formation and reactivation. Interestingly, we also observed a significant difference in engram size in the BLA of mice treated with minocycline compared to controls. Engram size has been shown to vary with learning, with

larger engrams supporting enhanced memory precision [70]. Relatedly, we previously reported a similar increase in DG engram size related to memory retrieval in MIA offspring, suggesting a common underlying mechanism [21].

These findings of larger engram underpinning memory retention appear somewhat contradictory to work by [71] who found that engram sparsity, namely in the CA1, is associated with the emergence of precise, adult-like memory. However, this may reflect that the memory retention we observe in minocycline-treated mice may not be due to an acceleration of memory maturation, but instead due to a prevention of the normal developmental trajectory that would otherwise lead to infantile amnesia. Supporting this interpretation, we observed a reduced number of PNNs in the DG and CA1 of minocycline-treated mice compared to controls. In contrast, [71] demonstrated a developmental increase in PNNs underpins the emergence of engram sparsity and memory precision (S2 Fig). Based on these findings, we propose that by inhibiting microglial activity during this developmental window, we prevent the normal trajectory of development that would otherwise lead to infantile amnesia.

Our previous work found that MIA in mice prevents infantile amnesia in male offspring. MIA is characterized by neuroinflammatory changes in the embryonic brain that can alter the trajectory of brain development and can also lead to persistent inflammatory changes in offspring postnatally and beyond into adulthood [20,72]. There is mounting evidence from rodent work that microglial function is perturbed by MIA and other prenatal environmental stressors, and this dysfunction may underpin many of the brain and developmental and behavioral abnormalities reported in MIA offspring [23,24,54,73]. This motivated us to investigate the potential relationship between MIA, microglia, and infantile amnesia given that we have observed changes in infant memory following both MIA or microglial inhibition. We found changes in some microglial-related genes in the embryonic brain following maternal Poly(I:C) injection, most notably a decrease in the microglial innate immune receptor TREM2 that governs microglial phagocytosis (S6 Fig). Interestingly, microglial TREM2 is reduced in human autistic patients and mice lacking TREM2 demonstrate altered brain connectivity and ASD-like behaviors [74]. Several rodent studies have targeted microglia in MIA offspring either through pharmacological inhibition with minocycline or complete depletion and have found that this leads to a prevention or reversal of some ASD behavioral deficits [22,57,75]. Our goal was to target microglia during an earlier postnatal window from P0-P14 (Fig 4A). Matcovitch-Natan and colleagues revealed that discrete developmental stages are underpinned by specific microglia transcriptomic profiles that likely reflect their respective developmental niche and MIA prematurely shifted microglia to a more "adult-like" state [58]. We hypothesized that inhibiting microglia during early postnatal development may realign them to their normal development stage permitting the emergence of normal behaviors such as infantile amnesia. The inhibition followed by removal may also result in a "rebound" effect of microglial activity, in a similar way to what has been suggested for depletion and repopulation that corrects microglial function [57]. In our investigation, we found that this early postnatal minocycline treatment prevented the persistence of infant fear memory in MIA offspring, likely through the modulation of microglia (Fig 4). This view is supported by our histological results which demonstrated that MIA resulted in a blunted microglia phenotype that was reversed by minocycline treatment (Fig 4G). Overall, our data is coherent with well existing evidence highlighting a crucial role for microglia dysfunction in MIA-induced behavioral deficits, including our novel memory phenotype of persistent infant engram expression. This result also strengthens our claim that microglia play an important role in infantile amnesia and additionally demonstrates important overlap between the developmental and cognitive changes in MIA offspring and microglia function. Yet, there is a need for future research to accurately define the phenotype and activity of microglia in MIA offspring during early postnatal development and how this contributes to changes in infant memory.

Given the emerging role of microglia in natural forgetting and the possibility that infantile amnesia represents an accelerated form of this process, it is plausible that microglial modulation of engram cells may act as a general mechanism of modulating engram expression. Under homeostatic conditions, baseline levels of microglial activity may govern natural forgetting, but during periods of elevated microglial activity, such as early postnatal development, may drive infantile amnesia. Furthermore, it is reasonable to suggest that disproportionate microglial activation could also result in maladaptive or pathological forgetting similar to what we observe in AD or TBI, while underactive microglia may give rise to

excessive memory recall leading to disorders such as post-traumatic stress disorder or the blunted microglia phenotype in MIA may explain the lack of infantile amnesia in those mice [76,77].

One possible cellular mechanism through which microglia drive infantile amnesia or forgetting in general is direct synaptic modulation through pruning or synaptic remodeling. This idea is supported by reports of complement-driven synaptic elimination underpinning forgetting in adult mice [43]. Microglia engage in high levels of synaptic pruning throughout early postnatal development that is crucial for establishing accurate connectivity, which makes this a plausible mechanism through which microglia mediate infantile amnesia [29,30,78]. Synaptic maturation and elimination also peak around the third postnatal week in mice, which coincides with the boundary of infantile amnesia [31]. We observed a decrease in CD68 expression following minocycline treatment which indicates a decreased level of phagocytic activity in microglia. In further support of synaptic pruning as a mechanism, a disruption in synaptic pruning and increased synaptic density has been reported in MIA offspring [22,23,79]. We previously reported an increase in spine density on DG engram cells of MIA offspring that could possibly be related to a reduction in synaptic pruning by microglia which as a result leads to the persistence of infant memories [21]. In addition to direct synaptic elimination, microglia can interact with and remodel ECM structures such as PNNs, that may indirectly influence synaptic plasticity and neuronal function and subsequently effect memory [80,81]. We found a significant difference in PNNs in DG and CA1 hippocampal subregions between minocycline and control-treated mice at P25 (S2 Fig). PNNs significantly increase and stabilize between P21 and P28 in the hippocampus [47,71]. Importantly, disrupted microglial homeostasis can lead to decrease PNNs, as has been observed in mice with Csfr1 haploinsufficiency [82]. Moreover, a decreased number of PNNs has been observed in MIA offspring [83–85]. A disruption of engram connectivity and memory accessibility may also occur due to alterations in the level of neurogenesis, and notably microglia are known regulators of neurogenesis [6,86–88]. Indeed, the neurogenic hypothesis of infantile amnesia suggests that high levels of postnatal neurogenesis lead to accelerated forgetting and is supported by work demonstrating that inhibiting neurogenesis prevents forgetting in infancy [6,89]. Similarly, an acceleration of neurogenesis in mice increases infant spatial amnesia [90]. This bi-directional effect of neurogenesis makes it a potential synergistic mechanism through which microglia modulate memory. This possibility resonates with findings of disrupted neurogenesis in offspring following prenatal immune challenge [91–93].

Finally, another possibility is that rather than impeding the target engram, microglia facilitate interference of competing engram ensembles. Interference describes a common type of forgetting where competing information impairs encoding or consolidation or retrieval of memory traces [94,95]. The presence of competing engram ensembles has been shown to interfere with original engram activation and memory recall, resulting in forgetting [95–97]. Microglia are tightly attuned to neuronal activity and can directly feedback onto neurons to control activity [33,98]. It is conceivable that microglia also modulate the activity of competing engram ensembles to promote their activation resulting in suppression of the original engram.

A limitation of the present study is the use of minocycline hydrochloride as a primary method of microglial manipulation. However, this was the most effective pharmacological agent as our attempts with Csfr1 inhibitors PLX5622 and BLZ945 at the required developmental stages either did not successfully deplete microglia or required diluents such DMSO which can have negative impacts on brain development and plasticity. We recognize that minocycline does not have specificity for microglia, and we cannot rule out the possible antimicrobial effects on the gut microbiome. However, we demonstrate that maternal and postnatal HFD, which is known to alter gut microbiome composition does not affect infantile amnesia (S5 Fig) [99,100]. Also, inhibition of microglial CX3CR1 results reproduced the same effect on infantile amnesia, giving us confidence in our findings. Additionally, tools for microglial manipulation remain limited, in particular for developmental studies, however, more recent developments in genetic manipulation may permit deeper investigation into the role of microglia in memory and forgetting in the future [101]. We acknowledge the importance of including female mice for sex-balanced experiences. In this case, we did not see any sex differences in our original experiments and chose to continue with males for our MIA questions as the prevention of infantile amnesia in MIA offspring is a male-specific

phenotype. Finally, in this investigation, we focused specifically on the role of microglia in infantile amnesia, but we do not discount the possibility of an involvement of other nonneuronal cells such as astrocytes which have also recently been implicated in memory and engram modulation [102].

## Conclusions

Infantile amnesia is possibly the most ubiquitous form of forgetting, and the non-pathological and highly conserved nature of this form of memory loss hints at its adaptive function. Rather than being an idiosyncratic case of memory loss, it may share similar mechanisms to other forms of forgetting and therefore using infantile amnesia as a platform, we may uncover more about memory management in the brain in general [103].

In conclusion, the findings outlined here reveal a novel role for microglia in infantile amnesia and demonstrate a mechanistic relationship between microglia, MIA, and infantile amnesia. Perturbation to microglia function in MIA or other neurodevelopmental insults may lead to altered development trajectories and changes information storage and retrieval. These findings support a growing body of evidence for a general role of microglia in memory management and thus microglia may represent an important target for memory and memory loss but also for neurodevelopmental alterations.

## Inclusion and diversity

We support inclusive, diverse, and equitable conduct of research.

## Methods

### Ethics statement

All experimental manipulations were carried out in accordance with Health Products Regulatory Authority (HPRA) Ireland guidelines and institutional policies on care and welfare of laboratory animals with ethical approval from the Trinity College Dublin Animal Research Ethics Committee and from the HPRA (Project Authorization Number AE19136/P166 and AE19136/P143).

### Experimental model and subject details

**Mouse model.** Two mouse lines were used for experimental purposes throughout this investigation. For wild-type experiments, the C57BL/6J substrain C57BL/6JOlaHsd mice were used. For engram labeling TRAP2-AI32 mice were used. The TRAP2-Ai32 line was generated by crossing Fostm2.1(icre/ERT2)Luo/J with Ai32(RCLChR2(H134R)/EYFP). Heterozygous mutants for transgenes CreERT and EYFP were used for all engram labeling experiments. Mouse genotypes were confirmed by sending tissue DNA samples to Transnetyx DNA extraction and genotyping.

All mice were bred in the animal facility at Trinity Biomedical Sciences Institute and maintained on a 12 h light/dark cycle. All experiments were conducted during the light phase. The day of birth was designated postnatal day 0 (P0). Infant mice (P17) were housed with mother until weaned (P21) and were then group housed 2–6 per cage in transparent plastic individually ventilated cages (GM500 cages, 501 cm$^2$ floor area) with nesting material, some enrichment in form of plastic tunnels and access to food and water *ad libitum.* For behavioral experiments, male mice were used unless otherwise stated. For infant experiments, each litter was counterbalanced across experimental groups where possible to limit litter effects. Where this was not possible, nested statistical analysis was performed per litter.

**Handling.** All behavioral experiments were conducted during the facility light cycle of the day (7am to 7 pm). All mice were habituated to handling prior to behavioral experiments. Mice were handled individually by the investigator for 3 min on 3 separate days with the final day of handling the day before the experiment. Infant mice were handled as a litter on the first day of handling (P14) to reduce anxiety. Mice were transported to and from the experimental room in separate Perspex cages.

**Engram labeling strategy.** To label memory engram cells, we used an inducible genetic labeling system using 4-hydroxytamoxifen (4-OHT) (Santa Cruz). 4-OHT was dissolved at 20 mg/mL in ethanol. The dissolved 4-OHT was aliquoted and stored at −20° for up to several weeks or used immediately. On the day of use, the dissolved 4-OHT was mixed with

Chen Oil (4 parts sunflower seed oil and 1 part castor oil) at a concentration of 10 mg/mL and vortexed. The ethanol was then evaporated by vacuum centrifugation. Mice were i.p injected with 4-OHT (50 mg/kg) immediately after the learning event. The TRAP2 line expresses the inducible c-*fos* promoter. Injection of 4-OHT activates iCre recombinase which when activated translocates to the nucleus and acts on two loxP sites removing the stop codon that otherwise prevents the expression of the ChR2/EYFP transgene in absence of 4-OHT. ChR2/EYFP transgene expression is driven by the pCAG promoter in iCre-expressing tissue 72 hours after 4-OHT injection. To evaluate activity-dependent expression of ChR2-EYFP, Kainic acid was dissolved in deionized water (5 mg/mL), aliquoted and stored until use. Mice were injected intraperitoneally with 20 mg/kg kainic acid followed by intraperitoneal injection of 4-OHT acid injection. Mice were sacrificed and immunohistochemically analyzed.

## Drug administration

**Minocycline.** Minocycline hydrochloride was obtained from sigma. Minocycline was dissolved in sterile saline to (10 mg/mL) and injected intraperitoneally (50 mg/kg) or given through drinking water. When administered through water, weight and water consumption were monitored closely and concentration of minocycline in water was adjusted accordingly. Water was replaced every 1–2 days.

**JMS-17-2.** JMS-17-2 was obtained from MedChemExpress. JMS-17-2 was dissolved in corn oil (1 mg/ml) and injected intraperitoneally (10 mg/kg).

## Maternal immune activation

Eight- to twelve-week old virgin female mice were paired with a male overnight for one night. The following morning the females were weighed and checked for seminal plugs before being returned to their homecage with cage mates. This was noted embryonic day 0.5 (E0.5). On E12.5 female mice were weighed again and pregnant dams were injected subcutaneously with a single dose of Poly(I:C) (20 mg/kg) or control (phosphate-buffered saline [PBS]). Poly(I:C) HMW (InvivoGen) was dissolved in sterile water at 67°C for 10 min and the aliquots are stored at −20° until use. Prior to use each aliquots were defrosted and injected subcutaneously (s.c; 20 mg/kg).

**Embryonic extractions.** Twenty-four hours after subcutaneous administration of either PBS or Poly (I:C) (20 mg/kg), dams were deeply anesthetized and embryos were extracted. Embryos were gently microdissected in RNase-free PBS to remove forebrain. Tissue was snap frozen in liquid nitrogen and stored at −80°C until further use.

**RNA extraction and qPCR.** Samples were homogenized in Trizol and collected into clean pre-labeled tubes. Twenty μl chloroform was added to each tube and tubes were agitated, left at room temperature for 2 min followed by centrifugation at 12 rcf for 15 min. The RNA was transferred to a new tube and 50 μl isopropanol was added to each tube to precipitate RNA. Tubes were inverted and left at room temperature for 10 min followed by centrifugation at 12 rcf for 10 min. The resulting supernatant was gently discarded and 100 μl 75% ethanol was added. This was centrifuged again at 7.5 rcf and supernatant was discarded and pellets were allowed to dry for 10 min or until the pellet turned transparent. Twenty μl RNAse-free water was added to pellet to resuspend. RNA was left on ice for 30 min followed by 15min in heat block at 55°. A nanodrop 2,000 UV spectrophotometer was used to assess RNA quality and concentration. Twenty μl cDNA was subsequently synthesized from 2 μg of isolated RNA using a cDNA reverse transcription kit (Qiagen QuantiTech RT kit) in a MiniAmp Thermal Cycler. qPCRs were then performed to quantify the relative mRNA expression of genes of interest. Gapdh mRNA quantification was used a control for normalization. Relative mRNA levels were calculated using the cycle threshold method. Primers are listed in Table 1.

## Behavior

**Context presentations.** Mice were exposed to two distinct contexts. Context A was a 31 × 24 × 21 cm Med Associates chamber with removable grid floor (bars 3.2 mm diameter spaced 7.9, apart), opaque triangular ceilings, and scented with

**Table 1. List of primers.**

|  | Forward sequence | Reverse sequence |
|---|---|---|
| **GAPDH** | 5'-GACGGCCGCATCTTCTTGT-3' | 5'-GACGGCCGCATCTTCTTGT-3' |
| **CX3CR1** | 5'-GTTGCCTCAACCCCTTTATCT-3' | 5'-CAGGAGAGACCCATCTCCC-3' |
| **TREM 2** | 5'-TGTGGTCAGAGGGCTGGACT-3' | 5'-CTCCGGGTCCAGTGAGGA-3' |
| **BDNF** | 5'-GCGGCAGATAAAAAGACTGC-3' | 5'-GCAGCCTTCCTTGGTGTAAC-3' |

0.25% benzaldehyde. Context B was a 29×25×22 cm, Coulbourne Instruments chamber with white floor and scented with 1% acetic acid. Chambers were cleaned with unscented Trigene before and after each mouse.

**Contextual fear conditioning.** Using a CFC paradigm mice were trained in Context A. Mice were place in context A for 6 min. After 3 min, three 0.75 mA foot shocks were delivered at 1-min intervals. Control groups (no shock) received no foot shocks but underwent the same contextual exposure. Mice were returned to home cage immediately after each session. Chambers were cleaned with unscented Trigene before and after each mouse.

Recall or generalization test sessions took place in Context A or B and conditions were identical to training, but no foot shock were delivered and each recall session lasted 3 min. Mice were returned to home cage immediately after each session. Chambers were cleaned with unscented Trigene before and after each mouse. Percent time each mouse spent freezing was quantified.

**Three-chamber social test.** Mice were habituated to the chamber with two empty holders for 10 min. The next day, mice were placed in the middle of the chamber and allowed to explore the three chambers for a period of 5 min. During this testing period, a social object (novel mouse) was contained in one holder in one chamber and an inanimate object (lego blocks) was contained in a holder in the other chamber. Time spent in each chamber, investigation time, and distance traveled were tracked using ANY-maze video tracking software. Investigation time consisted of the time the mouse head was oriented and within 10 mm of object or mouse.

**Marble burying.** A large testing arena 60×60×30 cm was filled 5 cm deep with wood chipping bedding, lightly packed down to make an even flat surface. A consistent pattern of 20 identical glass marbles (15 mm diameter) were evenly placed (4 cm apart) on the surface of the wood chip bedding. Mice were left alone in the testing arena for 30 min. A picture was taken before and after the test for analysis. A marble was considered buried if 2/3rds of the depth of the marble was buried.

**Analysis and statistics.** All behavioral experiments were analyzed blind to the experimental groups. All videos were randomized before manual scoring. Behavioral performance was recorded by digital video camera. CFC videos were manually scored individually. For the 3-chamber social test Any Maze software was used. Data analysis and statistics were conducted using GraphPad Prism 10 (GraphPad software). Unpaired Student's t-tests were used for independent group comparisons. Paired Student's t-tests were used to assess differences within groups. ANOVA followed by a Bonferroni *post hoc* test was used to determine conditions that were significant from each other where appropriate. Nested analysis was used where mice from the same litter could not be counterbalanced across experimental groups. Nonparametric tests were used where data was not normally distributed. All data were graphed as mean + Standard errors of the mean (SEM). An alpha level of 0.05 was used as a criterion for statistical significance.

## Immunohistochemistry

**Tissue collection.** Mice were sacrificed by overdose with 50 μl sodium-pentobarbital and perfused transcardially with PBS followed by 4% paraformaldehyde (PFA). Brains were then extracted and stored in 4% PFA overnight before being transferred to PBS for longer-term storage at 4°.

**Immunostaining.** Fifty μm coronal slices were cut using a vibratome and collected in PBS. Slices were washed and in PBS-Triton X-100 (PBS-T) 0.2% followed by a 1 h blocking in PBS-T with 10% normal goat serum at room temperature before being incubated with the primary antibody at 4°C overnight. On the following day, slices were washed in PBS-T

0.1% followed by an incubation with the secondary antibody before undergoing another round of washing using PBS-T 0.1%. Finally, slices were incubated for 10 min with DAPI antibody (1:1,000) to label cell nuclei before a final wash in PBS. Vectashield DAPI was used to mount the slices onto superfrost slides. Antibodies are listed in Table 2.

### Imaging and image analysis

To confirm expression and staining sections were visualized using Olympus BX51 upright microscope. For cell counting and microglial analysis, images were acquired using Leica SP8 gated stimulated emission depletion (STED) nanoscope. All images were taken at 40×.

**Microglia 3D reconstruction.** Fifty μm coronal slices were stained with anti-Iba1 and anti-CD68 or anti-GFP for 24 hours, followed by Alexa Fluor 488 and 568 conjugated secondary antibody and DAPI nuclear stain. Images were acquired using Leica SP8 gated STED nanoscope using a 40× oil objective and imaging parameters were kept consistent across all groups. Z stacking was performed with 0.5 μm steps in the Z direction through the entire 50 μm thickness, and 512 × 512-pixel resolution images were analyzed using IMARIS 9.6.2 software (Bitplane) (S3 Fig). Two to four ROI images were taken for each animal and 4–5 microglia were analyzed from each image. The IMARIS "surface" function was used to create a 3D surface rendering of the microglia and CD68 puncta. Both the area and volume of Iba1+ cells and CD68+ puncta were calculated. The IMARIS "filaments" function was used to trace and construct models of individual microglial cells and subsequently quantify filament length, branch points, and terminal points. Individual mice were used as independent samples.

For analysis of microglia-engram interactions, the "surface" function was used to reconstruct both Iba1+ microglial and YFP+ engram dendrites. The number of points of overlap between both surfaces was automatically quantified by assessing the number of points (distance ≤ 0 μm) between Iba1+ and YFP+ surfaces. The volume of overlapped surfaces was normalized by volume of YFP+ surface per image. A second normalization was carried out to normalize for microglia complexity. Individual mice were used as independent samples.

$$(Volume\ YFP+\ surface\ overlapping\ with\ Iba1+\ surface / Volume\ of\ YFP+\ surface)/ \\ (Average\ number\ of\ microglial\ branch\ points)$$

**Engram counting.** To quantify engram size and engram reactivation following contextual exposure, we quantified the number of labeled engram cells (EYFP+) and the extent of overlap between these cells and active cells (c-Fos+). Animals were euthanized and perfused 45 min after the behavioral assay. Sections were stained with anti-GFP and anti-c-Fos overnight followed by incubation with Alexa Fluor 488 and 548 conjugated secondary antibodies and DAPI nuclear stain. Images were acquired using Leica SP8 gated STED nanoscope using a 40× oil objective and imaging parameters were

**Table 2. Antibodies.**

| Primary antibodies | | |
|---|---|---|
| Anti-GFP chicken | Invitrogen | 1:1,000 |
| Anti-c-Fos rabbit | Synaptic Systems | 1:1,000 |
| Anti-Iba1 rabbit | Wako Fujifilm | 1:1,000 |
| Anti-CD68 rat | BIO-RAD | 1:500 |
| Lectin from *Wisteria Floribunda,* biotin conjugate | Sigma | 1:1,000 |
| **Secondary antibodies** | | |
| Antichicken Alexa 488 | Invitrogen | 1:500 |
| Antirabbit Alexa 568 | Invitrogen | 1:500 |
| Antirat Alexa 488 | Biosciences | 1:500 |
| Streptavidin, Alexa 647 conjugate | Thermofisher | 1:500 |

kept consistent across all groups. The brain area of interest was manually identified, and the area of each region was calculated using Fiji ImageJ software. The number of DAPI cells in DG was calculated first by taking the average diameter of a sample of DAPI+ cells in the DG from each animal and the area of the cells was calculated. The total number of cells in the DG was estimated by dividing the total area by the calculated cell area. For the BLA, CeA, and RSC, the number of DAPI cells in three randomly selected regions of interest were counted and used along with total area of the region to determine an estimate of the total number of DAPI cells. The number of EYFP+ and c-Fos+ cells were identified and counted manually using the Adobe Photoshop 2024 "count" tool.

We quantified the number of EYFP+ cells, the number of c-Fos+ cells, and the overlap (EYFP+c-Fos+) in the DG, BLA, CeA, and RSC. Any cell that was both EYFP+ and c-Fos+ was considered a "reactivated cell" and was both active during the original and recall experience. To calculate the percentage of cells expressing EYFP in each region, the number of EYFP+ cells was divided by the total number of DAPI cells. To calculate the percentage of cells expressing c-Fos in each region, the number of c-Fos+ cells was divided by the total number of DAPI cells. Engram reactivation was quantified in two ways: either by quantifying the number of double-positive EYFP+c-Fos+ cells as a percentage of total EYFP+ cells or total DAPI cells.

**Perineuronal net quantification.** For quantification of PNN animals were euthanized and perfused. Sections were stained with biotinylated Lectin from *Wisteria Floribunda* (WFA) overnight followed by incubation with Streptavidin, Alexa 647 conjugate and DAPI nuclear stain. Images were acquired using Leica SP8 gated STED nanoscope using a 40× oil objective and imaging parameters were kept consistent across all groups. The brain area of interest was manually identified, and the area of each region was calculated using Fiji ImageJ software and manual counting of WFA staining was also performed using ImageJ.

## Supporting information

**S1 Fig. Pharmacological inhibition of microglia prevents infantile amnesia. (A)** Schematic diagram of experimental schedule. Mice were injected with 50 mg/kg minocycline (mino) or control from P16-P25. **(B)** Schematic of behavioral schedule. Mice were trained at P17 and underwent recall test 1 day (Test 1) and 8 days (Test 3) post-training. Mice underwent a generalization test in Context B at P19. **(C)** Quantification of freezing behavior. **(D)** Schematic diagram of experimental schedule. Mice were injected with 50 mg/kg mino or control from P20-P25. **(B)** Schematic of behavioral schedule. Mice were trained at P17 (S) or received context exposure with no shock (NS) and underwent recall 8 days post-training. **(F)** Freezing behavior of mice that received either S or NS and mino treatment or control. **(I)** Comparison of freezing behavior between male and female mice treated with minocycline or control 1 and 8 days post-training. **(J)** Schematic diagram experimental schedule. Mice were administered JMS-17-2 (10 mg/kg) through i.p injection from P16-P25. **(K)** schematic of behavioral schedule. Mice were trained on P17 and tested for recall 8 days later. **(L)** Quantification of freezing behavior between shock (S) and no-shock (NS) JMS-treated mice. Black lightning symbol represents foot-shocks and empty lightning symbol indicates no-shock. Data is presented as mean ± SEM. ($n = 8$–11). (C, I) Statistical comparison performed using RM Two-way ANOVA with Bonferroni; (F) One-way ANOVA with Bonferroni; (L) Student's $t$ test; n.s $P > 0.05$, $*P < 0.05$, $**P < 0.005$. Details of all statistical comparisons may be found in S1 Data. The data underlying this Figure can be found in S2 Data.
(TIF)

**S2 Fig. Minocycline treatment inhibits the developmental increase in PNNs. (A)** Schematic diagrams of experimental schedule. Mice were administered minocycline (50 mg/kg) through drinking water from P16-P25. Following recall on P25 mice were sacrificed and transcardially perfused for brain tissue collection. **(B)** Representative images of hippocampal subfields analyzed for WFA staining. Scale bar = 150 μm. **(C–F)** Quantification of PNNs per area in DG, CA1, CA2, and CA3. **(G–H)** Representative images of WFA staining in controls and minocycline-treated mice. Scale bar = 75 μm. ($n = 4/5$).

Each point represents an individual mouse. Statistical comparison performed using Student $t$ test; n.s $P > 0.05$, *$P < 0.05$. Details of all statistical comparisons may be found in S1 Data. The data underlying this Figure can be found in S2 Data. (TIF)

**S3 Fig. Example microglia 3D reconstruction workflow. Step-by-step example of IMARIS 3D rendering and filament analysis. The IMARIS "surface" function was used to create surface of each channel of interest followed by selection and isolation of cells of interest**. The IMARIS 'filaments' function was used to trace and construct models of individual microglial cells and subsequently quantify filament length, branch points, and terminal points. For quantification of CD68 expression within microglial cells, the microglia surface was used to create a mask of CD68 channel and create a new surface of this masked channel. The volume of CD68 surface within each volume of microglia surface can be used to calculate relative CD68 expression. (TIF)

**S4 Fig. Postnatal minocycline treatment increases BLA reactivation only in mice that received a foot-shock.** **(A)** Schematic diagram of experimental schedule. Mice were administered 50 mg/kg minocycline from P16-P25 through drinking water. **(B)** Schematic of behavioral schedule. Syringe symbol represents 4-OHT injection. Black lightning symbol represents foot-shocks. **(C–F)** Quantification of % engram reactivation (EYFP+c-Fos+/DAPI) in control versus minocycline-treated mice. **(G, H)** Mice were trained at P17 and underwent either CFC (S) or Contextual exposure (NS) and mice underwent a recall test 8 days later. **(I–L)** Quantification of % Engram cells, % c-Fos, and engram overlap/reactivation in DG. **(M–P)** Quantification of % Engram cells, % c-Fos, and engram overlap/reactivation in AMG. $N = 5$–7 mice/group, $n = 4$ slices per mouse. Data is presented as mean ± SEM. Each point represents individual mice. Statistical comparison performed using Student's unpaired $t$ test; *$P < 0.05$. Details of all statistical comparisons may be found in S1 Data. The data underlying this Figure can be found in S2 Data. (TIF)

**S5 Fig. Maternal high-fat diet does not affect infantile amnesia. (A)** Schematic diagram of experimental schedule. Female mice are placed on a high-fat diet (HFD) 3 weeks prior to time-mating and remain on the diet through gestation and lactation until offspring are weaned at P21 while control mice remain on standard diet (SD). **(B)** Schematic of training and testing schedule. Male and female offspring underwent CFC at P17 and were tested for memory recall 8 days later. **(C)** Freezing behavior of mice during recall test. Data is presented as mean ± SEM. ($n = 6$–12) mice/group. Statistical comparison performed using Two-way ANOVA with Bonferroni; n.s $P > 0.05$. Details of all statistical comparisons may be found in S1 Data. The data underlying this Figure can be found in S2 Data. (TIF)

**S6 Fig. Changes in mRNA expression levels in embryonic brains following MIA and early postnatal minocycline treatment has no effect on WT mice. (A)** Experimental scheme. Pregnant female mice were injected with either PBS or Poly(I:C) at gestational day 12.5. Twenty-four hours following injection mice were deeply anaesthetized and embryos were extracted. Embryo brains were collected and snap frozen until use. RNA was extracted from brains, cDNA was generated, and qPCR was performed to quantify mRNA expression. **(B–D)** Relative expression of microglia-related mRNA between embryos from PBS or Poly(I:C) injected dams. *Created in BioRender. Stewart, E. (2025)* https://BioRender.com/pmb4ujv. (B) Relative mRNA expression of BDNF ($n = 13/17$). (C) Relative mRNA expression of CX3CR1 ($n = 6/11$). (D) Relative mRNA expression of TREM2 ($n = 10/11$). **(E)** Schematic diagram of experimental schedule. **(F)** Quantification of freezing in WT control mice treated with minocycline or vehicle from P0-P14. Data is presented as mean ± SEM. Statistical comparison performed using (B–D) Mann–Whitney test and (F) nest $t$ test. n.s $P > 0.05$. Details of all statistical comparisons may be found in S1 Data. The data underlying this Figure can be found in S2 Data. (TIF)

**S1 Data. Statistical summary.** A summary of all statistical tests performed and their outcomes. (XLSX)

**S2 Data. Data summary.** Excel file containing the raw data points for the data used in this manuscript. (XLSX)

## Acknowledgments

We thank Tamara Boto, David Loane, Michael-John Dolan, and past and present members of the Ryan Lab for scientific discussions and support. We thank Anna Connolly for support with histology experiments.

## Author contributions

**Conceptualization:** Erika Stewart, Tomás J. Ryan.

**Data curation:** Erika Stewart, Louisa G. Zielke, Gabrielle Guillaume Boulaire.

**Formal analysis:** Erika Stewart, Gabrielle Guillaume Boulaire, Tomás J. Ryan.

**Funding acquisition:** Tomás J. Ryan.

**Investigation:** Erika Stewart, Louisa G. Zielke, Antje R. de Boer, Gabrielle Guillaume Boulaire, Sarah D. Power.

**Methodology:** Erika Stewart, Sarah D. Power, Tomás J. Ryan.

**Project administration:** Erika Stewart, Tomás J. Ryan.

**Resources:** Erika Stewart, Tomás J. Ryan.

**Supervision:** Sarah D. Power, Tomás J. Ryan.

**Validation:** Erika Stewart, Antje R. de Boer, Tomás J. Ryan.

**Visualization:** Gabrielle Guillaume Boulaire.

**Writing – original draft:** Erika Stewart, Tomás J. Ryan.

**Writing – review & editing:** Erika Stewart, Louisa G. Zielke, Sarah D. Power, Tomás J. Ryan.

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
