## [Editor Report · Decision Letter 0]

9 May 2025

Dear Tomás,

Thank you for submitting your manuscript entitled "Microglial plasticity across development mediates infantile amnesia" for consideration as a Research Article by PLOS Biology.

Your manuscript has now been evaluated by the PLOS Biology editorial staff and I am writing to let you know that we would like to send your submission out for external peer review.

Once your full submission is complete, your paper will undergo a series of checks in preparation for peer review. After your manuscript has passed the checks it will be sent out for review. To provide the metadata for your submission, please Login to Editorial Manager (https://www.editorialmanager.com/pbiology) within two working days, i.e. by May 13 2025 11:59PM.

Kind regards,

Christian

Christian Schnell, PhD

Senior Editor

PLOS Biology

cschnell@plos.org

---

## [Decision Letter · Decision Letter 1]

16 Jun 2025

Dear Tomás,

Thank you for your patience while your manuscript "Microglial plasticity across development mediates infantile amnesia" was peer-reviewed at PLOS Biology. It has now been evaluated by the PLOS Biology editors, an Academic Editor with relevant expertise, and by several independent reviewers.

In light of the reviews, which you will find at the end of this email, we would like to invite you to revise the work to thoroughly address the reviewers' reports.

As you will see below, the reviewers that the reviewers think that the study is very well executed and provides important insights. Reviewer 1 requests a few methodological clarifications and an additional control experiment, while Reviewer 2 asks you to show more of the raw data and expand the discussion. Reviewer 3, however, identified a few internal inconsistencies and mechanistic gaps that likely require additional experimental data.

Given the extent of revision needed, we cannot make a decision about publication until we have seen the revised manuscript and your response to the reviewers' comments. Your revised manuscript is likely to be sent for further evaluation by all or a subset of the reviewers.

We expect to receive your revised manuscript within 3 months. We understand that these revision may need a bit more time, so please email us (plosbiology@plos.org) if you have any questions or concerns, or would like to request an extension.

**IMPORTANT - SUBMITTING YOUR REVISION**

*Re-submission Checklist*

*Published Peer Review*

*PLOS Data Policy*

*Blot and Gel Data Policy*

Sincerely,

Christian

Christian Schnell, PhD

Senior Editor

PLOS Biology

cschnell@plos.org

REVIEWS:

Reviewer #1: Summary: This study reveals a novel role for microglia in regulating memory accessibility during early life, offering new insight into the mechanisms of infantile amnesia. By profiling microglial changes across postnatal development, the authors show that inhibiting microglial activity prevents infantile forgetting of a contextual fear memory. Using activity-dependent tagging, they demonstrate that microglial inhibition alters engram size, reactivation, and interactions in the amygdala. Notably, microglial dysfunction was linked to persistent early-life memory in maternal immune activation (MIA) offspring. These findings position microglia as active modulators of infant memory and highlight their relevance in neurodevelopmental disorders. There are a few areas of improvement that should be addressed prior to publication.

Major Concerns:

1. For Figure 1 B and C, it would be helpful to show the initial training responses to each shock. How were the shock intensities chosen? Were they the same for the two ages? Ideally, the immediate shock response and freezing would be the same at training across age. Please also make it clear if the same animals were tested repeated across days or different animals were tested for each day. If the same animals were retested, how was extinction accounted for? If different animals were tested, please remove the dotted line between timepoints as it implies a repeated test of the same animals. It is also a bit confusing to use postnatal day for B and time post testing for C.

2. Only 3-5 microglia were analyzed per brain region per mouse. How were these microglia selected? Was the experimenter blinded to treatment condition? How was it determined that only 3-5 cells was representative of the microglia within an entire region given the large and well documented heterogeneity in microglia? Same comment for the microglial co-localization with CD68 analysis.

3. For all figures, please specifically denote males and females with different dot colors or shapes where both sexes were used. Clearly denote when just male mice were used. Where both sexes are used, sex should be included in the statistical analysis as a main effect and interaction. If sex was not included in the majority of the mechanistic studies, consider adding females for a subset of measures or discussing this as a limitation in the discussion and conclusions.

4. There are several instances of the use of the phrase "microglial activation" (ie. line 203, 211 etc). It is unclear what exactly this means. The changes in morphology and contact with neurons suggest changes in microglia activity, which is not the same thing as activation, which in the literature implies immune activity or inflammation. Please clarify the exact meaning of activation and consider other word choices for clarity.

5. It is surprising that the Mino group showed reduced microglia-neuron contacts (Figure 3Z) while also showing more branching (3U-W). Were there any changes in the neuronal structure between groups that could help explain this?

6. It is unclear how MIN can both facilitate amnesia (Figure 4B) in MIA and prevent amnesia in WT mice (Figure 2C). Similarly, in MIA MINO increases CD68 in microglia (Figure 4G) and in WT MINO decreases it (Figure 2E). One obvious difference is the timing of MINO treatment between the two experiments. Please add a WT control condition to Figure 4 to clarify the impact of MINO on VEH treated animals under the same conditions as MIA.

Minor Concerns:

1. Line 66: "…microglia are of the one candidate mechanisms…" should be "microglia are one of the candidate mechanisms"

2. The text on lines 253-259 describes Figure 4H. There is no H in figure 4.

3. Line352: crucial for establish accurate connectivity should be establishing

4. Why was the RSC only included in the Figure 3 work and not in Figure 1?

5. Figure S2 is not really connected to the rest of the paper. Figure S4 is not cited in the text and Figure S3 is not cited until the conclusions (out of order and missing from the results).

Reviewer #2 (Alonso Martínez Canabal): The manuscript presented by Stewart et al. investigates the relationship between microglial activity and infantile amnesia—a characteristic of altricial mammals, including humans and many of the common rodent models used in biomedical research (though not all). This research is important and relevant because, despite the critical role of infantile amnesia in human cognitive and mental development, it has been historically neglected in neuroscience, particularly within the biomedical framework. While clinical psychology has paid somewhat more attention to the topic, the overall attention remains limited. This neglect has hindered our understanding of how infantile amnesia may relate to the development of neurodevelopmental disorders such as ASD and ADHD, as well as pediatric psychiatric conditions like early-onset depression or PTSD.

This study is also important for highlighting the role of microglia—a cell type often overlooked in cognitive neuroscience, which traditionally focuses on neurons and, to a lesser extent, astrocytes and oligodendrocytes. However, emerging research suggests that microglia play a more active role in brain development and circuit tuning than previously appreciated. The present study supports this notion by demonstrating a causal relationship between microglial activation and the persistence of memory formed during the preweaning period and tested shortly after weaning.

The introduction is well-written, and the Materials and Methods section includes sufficient detail for replication by researchers with appropriate technical expertise. However, there is an inconsistency in the Results section: on line 132, the text refers to a graph (freezing behavior), but the corresponding figure shows a microphotograph. This should be corrected.

The subsection "Inhibition of microglial activity alters infant engram dynamics" is particularly compelling and central to the study. It is intriguing that engram activation in the dentate gyrus (DG) does not show significant differences, which contrasts with findings from Guskjolen et al. (2018), where this was a major result. The authors attribute their main findings to the amygdala; however, they do not present the corresponding behavioral data. Although the experimental design includes contextual fear conditioning (CFC) training and recall, freezing behavior is not clearly reported—except in Panel X as part of a Pearson correlation with endpoint measurements. It is essential to present the behavioral data (as done in Figure 2C for Mino vs. control groups) to fully understand the relevance of the observed engram activity.

In the Discussion, the authors cite Akers, Guskjolen, and Travaglia, who address both infant neurogenesis and electrophysiological maturation. It would also be appropriate to consider the work of López-Oropeza (2022). Despite these references, the discussion does not address the potential role of newborn neurons—known to be produced at a rate approximately seven times higher in infants than in adults—in the context of infantile amnesia. Given that previous neurobiological hypotheses have proposed a key role for these neurons, this omission should be rectified. Furthermore, while the paper briefly mentions pruning, it does not explore the potential interplay between microglia and newborn neurons, which could be central to the mechanisms underlying infantile amnesia. The authors should expand the discussion to consider how microglial activity may interact with increased neurogenesis during infancy, particularly in light of the literature already cited in the introduction.

Reviewer #3: The authors investigate the role of microglia in episodic-like memory development by assessing morphometric changes and the effect of inhibiting their activity early in life on the expression of infantile amnesia and engram related activity. They observe that microglia may be causally involved in infantile amnesia, suggesting microglia-related maturational processes may be critical for healthy memory development, they argue their results has implications for our understanding of the cognitive deficits observed in maternal immune activation models.

The authors have done an impressive amount of work and the results would surely be of interest to the learning and memory neuroscience community as well as developmental researchers. That being said, I have a number of reservations regarding the morphological analyses carried out, the link between the different findings explained and how the results relate and advance the current state of the art. I detail specific points below.

* The first section of the paper describes developmental morphological changes in microglia. The authors note significant changes in DG microglial processes between P20 and P25 (and between P20 and P22 in AMG) but also find changes in adult samples when microglia processes are compared between animals whose memory was tested at different retention intervals. This begs the question, do the developmental changes relate to development or the longer retention interval in the older animals? It is difficult to answer this question with the current data. However, what the authors could analyse is how the different morphological markers change with the temporal extent of the retention interval (e.g. the adults show a reduction in branch points for more remote memories, is a similar temporal trend observed in development?). This would highlight any developmental changes. Further, they could comment on how the morphological markers differ in general between the pup and adult data. Looking at Fig1, it seems all the morphological measures are lower in quantity in development. However, this point is not discussed.

* The link between this first section is only superficially linked to the subsequent section(s). Namely, the subsequent sections effectively show that inhibiting microglia early in life leads to accelerated memory development (memories formed at P17 are retained at P25), suggesting perhaps microglia hyperactivity early in life may play a causal role in infantile amnesia, and thereby healthy memory development. Yet, the first section does not address overall activity of microglia in development, rather it focuses on specific morphological features. This makes it difficult for the reader to link the different section and to understand what the mechanistic link between microglia maturation and memory maturation is. Could the authors simply quantify microglia activity in development? This would help create more coherence between the different sections and help clarify the take-home message. Further, it would be useful if the authors used the discussion to synthesise better their different experimental findings.

* The second results section of the paper shows inhibiting microglial activity (via minocycline administration) prevents infantile amnesia. This is an interesting and novel result. However, the authors found paradoxically that PNNs were sparser in the minocycline treated mice. This results seems to contradict a paper published a couple of years ago (Ramsaran et al. (2023)) showing PNNs increase in expression in development and this increase is associated with memory maturation. Here, the authors essentially accelerate memory development but when they do this they see a reduction in PNNs. Could the authors please try to address this inconsistency. Do the authors think the mechanisms that underlie the precocious emergence of memory in the minocycline mice is mediated by different neurobiological processes than natural memory development?

* Moreover, the authors report that minocycline leads to an increase in engram size in the DG and AMG and conclude that the observation that engrams are similarly affected in different regions points to common mechanism behind infantile forgetting. Again, this seems somewhat contradictory to past work (Ramsaran et al. (2023)), where the developmental of sparse engrams is thought to drive memory development. The authors should try to reconcile these divergent results.

* The final results section attempts to relate the experimental findings in healthy developing mice to understand cognitive deficits in MIA mice. Here, the authors show that minocycline reinstates IA in MIA mice (that "naturally" do not show IA). However, this result does not seem logically consistent with the previous data. Namely, if MIA leads to reduced microglial activity in developing mice which prevents IA, then why would administering a drug that reduces microglial activity further lead to the emergence of IA in these mice? I believe the authors try to address this point in their discussion, but the logic of using minocycline to understand the neurobiological mechanisms that prevent IA in MIA needs to be explicitly explained in the results section to improve coherence and readability.

---

## [Decision Letter · Decision Letter 2]

30 Oct 2025

Dear Dr Ryan,

Thank you for your patience while we considered your revised manuscript "Microglial plasticity across development mediates infantile amnesia" for publication as a Research Article at PLOS Biology. This revised version of your manuscript has been evaluated by the PLOS Biology editors, the Academic Editor and one of the original reviewers.

Based on the reviews and on our Academic Editor's assessment of your revision, we are likely to accept this manuscript for publication, provided you satisfactorily address the remaining points raised by Reviewer 3. Please also make sure to address the following data and other policy-related requests.

1) We routinely suggest changes to titles to ensure maximum accessibility for a broad, non-specialist readership, and to ensure they reflect the contents of the paper. In this case, we would suggest a minor edit to the title, as follows. Please ensure you change both the manuscript file and the online submission system, as they need to match for final acceptance:

"Microglial activity during postnatal development is required for infantile amnesia in mice."

2) The Ethics statement needs to be a separate, independent (and the first) subheading in the Material & Methods section. It must include the full name of the IACUC/ethics committee that reviewed and approved the animal care and use, as well as the protocol/permit/project license number. https://journals.plos.org/plosbiology/s/ethical-publishing-practice

3) In Methods, please change the subheading "Subjects" to "Mouse model"

4) In the Introduction and Discussion, please be clearer when you are refering to mice vs humans

5) Please note that per journal policy, the model system/species (mice) studied should be clearly stated in the abstract of your manuscript.

6) Thank you for providing all raw and numerical data underlying the figures. While taking a look at the file we noticed a few labels mark wrong and will require for you to take a second look:

- 2E and 2G

- labels for Figures 3 and 4, might be swaped.

7) Please cite the location of the data clearly in all relevant main and supplementary Figure legends, e.g. “The data underlying this Figure can be found in S1 Data” or “The data underlying this Figure can be found in https://doi.org/10.5281/zenodo.XXXXX”

8) Supplementary files (e.g., excel). Please ensure that all data files are uploaded as 'Supporting Information' and are invariably referred to (in the manuscript, figure legends, and the Description field when uploading your files) using the following format verbatim: S1 Data, S2 Data, etc. Multiple panels of a single or even several figures can be included as multiple sheets in one excel file that is saved using exactly the following convention: S1_Data.xlsx (using an underscore).

9) Please make sure to deposition in a publicly available repository. Please also provide the accession code or a reviewer link so that we may view your data before publication. All raw data should be available to the public and not only upon request.

We expect to receive your revised manuscript within two weeks.

*Published Peer Review History*

*Press*

Sincerely,

Melissa

Melissa Vázquez Hernández, PhD

Associate Editor

PLOS Biology

on behalf of

Christian

Christian Schnell, PhD,

Senior Editor

cschnell@plos.org

PLOS Biology

REVIEWER'S COMMENTS

Reviewer #3:

I am generally happy with the edits the authors have made to the manuscript and their reply to my comments. I do, however, think it would be useful for the reader if the authors discussed the numerous "apparent" discrepancies between their paper and the Ramsaran (2023) paper, which I mentioned in my initial review. Namely, Ramsaran et al also studied engram size and changes in PNNs in relation to memory development. I recognise the memory processes and brain regions studied by Ramsaran et al differ from the current study, but I think it's worth discussing these so this does not cause friction or confusion in the field.

Otherwise I commend the authors for their work.

---

## [Editor Report · Decision Letter 3]

14 Nov 2025

Dear Tomás,

Thank you for the submission of your revised Research Article "Microglial activity during postnatal development is required for infantile amnesia in mice" for publication in PLOS Biology. On behalf of my colleagues and the Academic Editor, Jozsef Csicsvari, I am pleased to say that we can in principle accept your manuscript for publication, provided you address any remaining formatting and reporting issues. These will be detailed in an email you should receive within 2-3 business days from our colleagues in the journal operations team; no action is required from you until then. Please note that we will not be able to formally accept your manuscript and schedule it for publication until you have completed any requested changes.

PRESS

We frequently collaborate with press offices. If your institution or institutions have a press office, please notify them about your upcoming paper at this point, to enable them to help maximize its impact. If the press office is planning to promote your findings, we would be grateful if they could coordinate with biologypress@plos.org. If you have previously opted in to the early version process, we ask that you notify us immediately of any press plans so that we may opt out on your behalf.

Sincerely, 

Christian

Christian Schnell, PhD

Senior Editor

PLOS Biology

cschnell@plos.org